# Prevalence and Associated Factors of Depression in Medical Students in a Northern Thailand University: A Cross-Sectional Study

**DOI:** 10.3390/healthcare10030488

**Published:** 2022-03-07

**Authors:** Suwatthanachai Phomprasith, Nuntaporn Karawekpanyawong, Kanokporn Pinyopornpanish, Wichuda Jiraporncharoen, Benchalak Maneeton, Phichayut Phinyo, Suppachai Lawanaskol

**Affiliations:** 1Department of Psychiatry, Faculty of Medicine, Chiang Mai University, Chiang Mai 50200, Thailand; suwatthanachai_phom@cmu.ac.th (S.P.); nuntaporn.karawek@cmu.ac.th (N.K.); benchalak.maneeton@cmu.ac.th (B.M.); 2Department of Family Medicine, Faculty of Medicine, Chiang Mai University, Chiang Mai 50200, Thailand; knp_02@hotmail.com (K.P.); wichudaj131@gmail.com (W.J.); phichayutphinyo@gmail.com (P.P.); 3Center for Clinical Epidemiology and Clinical Statistics, Faculty of Medicine, Chiang Mai University, Chiang Mai 50200, Thailand; 4Chaiprakarn Hospital, Chiang Mai 50320, Thailand

**Keywords:** depression, well-being, loneliness, treatment continuation, emotional quotient

## Abstract

This study was conducted to investigate the prevalence and associated factors of depression in medical students. This cross-sectional study investigated the prevalence and associated factors of depression in medical students from May 2018 to April 2019. Depression was diagnosed using the nine-item Patient Health Questionnaire. We evaluated the following potential predictors: demographic data, stressors, psychiatric comorbidities, emotional intelligence (EI), and perceived social support. The association between potential factors and depression was analyzed using multiple logistic regression analysis. The prevalence of depression was 149 of 706 students with 12.5% suicidality. Second- and fourth-year medical students were high-risk groups. Risk factors identified were insufficient income, physical illness, and previous psychiatric illness. Depression in medical students likely coincides with anxiety, internet addiction, sleep problems, and loneliness. Highly associated stressors were personal relationships, physical health, mental health, difficulties in social relationships, satisfaction with grades, and boredom with medical education. Protective EI factors included emotional self-control, problem-solving abilities, inner peace, and life satisfaction. Up to 21.1% of medical students had depression. In this study, among multiple known risk factors of depression, we found that EI is the novel protective factor against depression among medical students. EI training might be protective intervention for medical students in the future.

## 1. Introduction

Depression can be characterized by persistent and recurrent sadness or lack of pleasure that eventually impairs a person’s functional ability to work, learn, or cope with daily life. The World Health Organization reports that the prevalence of depression worldwide is currently 4.4%. However, the prevalence of depression in medical students is reported to be 3–10 times higher (11.5–48.2%) [1,2,3]. Among Thai medical students, the prevalence of depression is 27% [4,5,6,7,8]. Depression is the most concerning mental health problem in medical students because it impacts students’ daily life, academic performance, and patient care [9,10]. Therefore, a comprehensive understanding of its impact and association is necessary to design preventive interventions. A study from Hawaii proposed the preventive intervention for depression, which consisted of faculty education, medical student counseling, and medical student education [11]. In Thailand, only screening and counseling is conducted to manage depression among medical students. Furthermore, no program for preventing depression has been established in Thailand. Exploring factors associated with depression among Thai medical students, especially protective factors, is beneficial for developing such a program.

Previous research has found an association between depression and relationship problems, lack of support systems, poor sleep quality, lack of motivation to study medicine, and pessimism [4,8]. Moreover, a Thai study in Vajira Hospital found an association between psychological abuse and depression [8].

Furthermore, many studies have shown that internet and game addiction have direct and indirect influences on anxiety, stress, and depression [12,13]. In addition, other psychiatric problems (anxiety, loneliness, and burnout) have been positively correlated with depression in many recent studies [14,15,16,17,18]. Conversely, emotional intelligence (EI) and perceived social support (PSS) are inversely correlated with depression and anxiety [19,20,21,22,23]. To our knowledge, these associated factors may also have correlations. Thus, adjusting for all factors that influence depression is necessary. Although many studies indicate associations between the said factors and depression, no study has suggested methods of adjusting all the potential factors of depression due to insufficient study size.

We hypothesized that a younger age (implying lower maturity) indicates worse mental health. Thus, an evaluation of depression throughout the years of study among Thai medical students is required. However, data on depression across the six years of medical school in Thailand are limited, especially in northern Thailand. Only one study was conducted across all year levels [8], and almost all studies in Thailand have been conducted among medical students in various levels of education [4,5,6,7]. Furthermore, studies of factors associated with depression in Thailand reported only problems but not protective factors.

For this reason, our primary objective was to investigate the prevalence of depression among northern Thai medical students. The secondary objective was to explore both potential risk and protective factors of depression across all year levels in medical school.

## 2. Materials and Methods

### 2.1. Study Design

This cross-sectional study was conducted at the Faculty of Medicine of Chiang Mai University in northern Thailand. The Research Ethics Committee of the Faculty of Medicine approved the study protocol (Reference Number 100/7003).

### 2.2. Research Setting

The Thai medical curriculum involves a six-year course and requires high school graduation for admission. The first-year curriculum consists of basic science lectures. Second- and third-year curricula involve a deeper study of medicine with lectures and problem-based learning of organ systems-based blocks. The last three years of medical school are the clinical training period.

### 2.3. Participants

We evaluated 1444 medical students of all year levels from May 2018 to April 2019. We contemporarily collect the data within a year across all year levels in medical school. Participants were asked to provide written consent and privately complete the anonymous self-reported questionnaires without receiving any compensation. Participants could withdraw at any time without giving a reason. Participants were limited to one response per person. The inclusion criteria of this study were current medical students, willing to participate, and having any device to conduct our online questionnaires. This study included both diagnosed and undiagnosed participants, providing a practical assessment of possible risk factors in a diverse population. We excluded students who did not provide consent or submitted incomplete questionnaires from further analysis.

### 2.4. Assessment of Potential Determinants for Depression

#### 2.4.1. General Demographic Data

The questionnaires consisted of items on sociodemographic and health-related characteristics, including sex, age, year of study [8], hometown [4,24,25], relationship status [26], financial status [27], accommodations [28], underlying diseases (including mental health difficulties and substance usage), and family psychiatric history. We also collected information on perceived causes of stress, such as education, finance, relationships, health, and others. The students were further interviewed about their social adjustment ability, academic satisfaction, and reasons for studying medicine.

#### 2.4.2. Psychiatric Problems

Anxiety: Generalized Anxiety Disorder-7 is a seven-item test. Anxiety was defined in this study as having a score of 10 or greater, with 89% sensitivity and 82% specificity [29].Burnout: The Thai version of the Maslach Burnout Inventory–Student Survey is a 15-item questionnaire. Burnout status was determined as having high emotional exhaustion (>14 points), high depersonalization (>6 points), and low professional efficacy (<12 points) (Cronbach’s alpha of 0.79) [30].Game addiction: The student version of the Game Addiction Screening Test has 16 items that assess problematic gaming behaviors in the past 3 months [31]. The cutoff for problematic game-playing was 24 points for male participants and 16 points for female participants (Cronbach’s alpha of 0.94).Internet addiction: The Thai version of the 20-item Internet Addiction Test assesses internet compulsivity and addiction. The cutoff for internet addiction is >30 points (Cronbach’s alpha 0.89) [32,33].Poor sleep quality: The Thai version of the Pittsburgh Sleep Quality Index (PSQI) is composed of seven components and evaluates sleep quality in the past month. Each component point ranges from 0 (no difficulty) to 3 (severe difficulty). A total PSQI > 5 was defined as poor sleep quality (Cronbach’s alpha 0.84) [34].Loneliness: The six-item revised University of California–Los Angeles (UCLA) Loneliness Scale evaluates feelings of loneliness during the past week. The higher the score, the higher the feelings of loneliness (Cronbach’s alpha 0.83) [35].

#### 2.4.3. Possible Protective Factors

Emotional intelligence (EI): The EI test for the Thai population aged 12–60 years was developed by the Mental Health Department of the Thai Ministry of Health. This 52-item self-report questionnaire consists of nine subscales. The interpretation for normal EI based on age for each subscale is presented in Appendix A
Table A1 (overall Cronbach’s alpha 0.85) [36,37].The revised Thai version of the Multi-Dimensional Scales of PSS (r-MSPSS) is a 12-item self-report measuring an individual’s experience of being supported by family, friends, and special persons. The higher the score, the higher the PSS level (Cronbach’s alpha of 0.91) [38].

#### 2.4.4. Depression

The depression group was determined using the nine-item Patient Health Questionnaire (PHQ-9, Thai version), which evaluated the frequency of depressive symptoms occurring in the past 2 weeks. The recommended cutoff score is ≥9 for diagnosing major depression with a sensitivity of 0.84 and a specificity of 0.77 (compared with the Thai version of the Mini International Neuropsychiatric Interview) [39]. The severity of depression is further classified into three groups: mild (9–14 points), moderate (15–19 points), and severe (≥20 points) [40].

### 2.5. Study Size Estimation

In the previous study, the prevalence of depression among Thai medical students was 30.54% [8,12]. We confirmed the adequacy of the statistical power of the study size using W.G. Cochran’s equation [41], obtaining an alpha of 0.05 and an error of 0.05%. Thus, 326 participants were required to obtain adequate statistical power.

### 2.6. Statistical Analysis

Categorical data were expressed as frequency and percentage. Normally distributed continuous data were expressed as mean ± standard deviation (SD). Non-normally distributed continuous data were expressed as median and interquartile range. Univariable logistic regression was used in the effect estimation of each predictor.

Regarding the exploratory model strategy, we included all known risk factors of depression in a multivariable analysis without pre-selection from the preceding univariable analysis. According to enough events per variable (EPVs), at least 10 EPVs were required for multivariable binary logistic regression [42]. However, the prevalence of depression was less than 10 times the number of the related factor; thus, including all factors into a single multivariable model is unacceptable. Therefore, we adjusted this effect using the confounder summary score concept [43].

First, we divided all potential risk factors into five domains (demographic data, stressors, psychiatric problems, EI domain, and social support). Each predictor domain was analyzed using multivariable binary logistic regression to estimate the probability of depression. Five models were generated, one from each domain. Each domain was influenced by the others. The predicted values from each model containing known potential factors from the first step were included in the final model as regressors. The investigators performed statistical analysis using the STATA program version 16 (StataCorp. 2019. Stata Statistical Software: Release 16. College Station, TX, USA: StataCorp LLC.). The statistical significance was *p* < 0.05 (two-sided).

## 3. Results

The overall response rate was 48.9% (706 of 1444), including 77.7% (195 of 251) of first-year medical students, 45.7% (217 of 475) of late second-year students, and 40.9% (294 of 718) of third- to sixth-year students (clinical training period). The percentages of female medical students and of those who repeated a grade were similar between participants and total population (female: 48.7% of 48.9%; repeated a grade: 3.3% of 3.3%).

The mean age of the participants was 20.6 ± 2.0 years old, with no significant difference between the depression and non-depression groups. Among participants with depression, no significant difference in depression was found among those who received a higher income per month. However, a significant difference in depression was reported among those who received insufficient income. The depression group had significantly more medical comorbidities, psychiatric disorders, and family histories of psychiatric disorder. Although most students in the two groups drank alcohol less than once a week and did not exceed a standard measurement drink, more students in the depressed group reported using alcohol as self-medication to reduce stress and improve sleep. Only 4.7% of participants smoked (≤10 cigarettes per day), with no significant difference in smoking between the depression and non-depression groups. No significant differences were found in other demographic data (Table 1).

### Depression

The prevalence of depression in this study was 21.1% (149 of 706). The most affected group were second- and fourth-year students. However, no significant differences were found among the year levels (*p* = 0.316). The rate of suicidality was 12.5% (88 of 706), with 62.5% (55 of 88) of students with suicidal ideation having some degree of depression. Sixth-year medical students had the highest rate of suicidality (Table 2).

Surprisingly, less than 20% (24 of 149) of participants in the depression group had ever made use of any mental health care service. About 70% (16 of 24) of them had been diagnosed with major depressive disorder (MDD); unfortunately, only 50% (7 of 16) of students with MDD continued treatment until the end of the study period.

All psychiatric problems and possible protective factors of interest were highly significant in univariable analysis (Table 3). Factors associated with depression from the five separate models using the confounder summary score model are shown in Table 2. Three demographic factors, six stressors, four psychiatric problems, and four EI domains were associated with depression. Confounder summary score was developed as shown in Table 4.

## 4. Discussion

This study found that the prevalence of depression in northern Thailand medical students was 21.1%, of which 12.5% had suicidal ideation. Furthermore, nearly 60% of sixth-year students who were depressed had suicidal ideation. Psychological problems associated with depression were anxiety, sleep problems, internet addiction, and loneliness. A possible protective factor against depression was EI, especially with regards problem-solving, life satisfaction, peace, and emotional self-control. Among medical students with depression, the significant stressors were mental health, relationships with loved ones, physical health, dissatisfaction with their course, and difficulty in social relationships. However, their satisfaction with grades had a positive effect on reducing depression. Demographic data related to depression included a history of psychiatric disorders, medical comorbidities, and sufficient income.

### 4.1. Prevalence

Medical students are undisputedly at a high risk of depression [3]. Prevalence of depression among Thai medical students is lower than the global rate (16.0–39.1% vs. 11.5–48.2%) [6,9,10,11,12], as confirmed in the present study. A higher prevalence of depression among second- and fourth-year medical students highlights the importance of the transitional periods in the course. The transition from learning basic science in the first year to learning medical concepts in the second year was frequently described as the most stressful transition due to changes in the learning environment, teaching styles, increased study burden, and competition among medical students. Moreover, the transition from the preclinical years to the first clinical year was the second most stressful transition. Many students reported experiencing a sense of lacking enough time and feelings of uselessness and inadequacy [44].

Our study emphasized the undertreatment of depression, which has remained unsolved for decades [28]. Medical students, whose teachers are doctors, nevertheless reported obstacles to accessing mental health care, usually due to a lack of time, stigmatization from themselves or others, and the fear of not achieving good grades and sabotaging their future career. More proactive solutions are required, which includes an increased role of the medical professors, not just the psychiatrists. In our opinion, medical professors have essential roles in solving these problems in many ways, for example:Providing time in the curriculum for students to take care of themselves;Identifying students highly prone to depression and providing sources of help;Taking a stand to reduce the stigmatization of depression; andUsing positive communication, such as humor [45].

### 4.2. Factors Associated with Depression in Medical Students

#### 4.2.1. Significant Psychiatric Problems

The significant association between anxiety and depression among medical students is not surprising, but the screening and management plan is. Anxiety co-occurs with depression frequently because of similar etiologies, such as genetic vulnerability, neuroticism, and elevated corticotropin-releasing factors [46,47]. In addition, anxiety increases the risk for depression due to increased impairment and debilitating factors [48]. Anxiety and depression, when combined, are more severe, more disabling, more resistant to treatment, and presents a greater risk of suicide than either disorder alone [49].

Robust research has shown sleep problems are a risk factor and symptom of depression [50]. Sleep problems involve a similar pathophysiology to depression, such as increased inflammatory cytokine expression, alterations in the level of monoamines (serotonin and norepinephrine), and interruption of the circadian rhythm. A genetic overlap between insomnia and depression has also been reported [51]. Moreover, the co-occurrence of sleep disturbance and depression decreases treatment response and increases depression severity, duration, and relapse rates [52,53].

We found that internet addiction was highly associated with depression. Depressed students tended to use the internet for social support, achievement, and pleasure of control, which can alleviate depressed feelings. However, at the same time, excessive internet use is a risk factor for depression because online communication displaces actual social communication, resulting in an ineffective ability to cope with difficult real-life situations [54].

Consistent with previous research [55], this study found an association between a high loneliness score with depression among medical students. Therefore, we hypothesized that we could reduce depression by reducing loneliness. To our knowledge, four primary intervention strategies for reducing loneliness have been identified: improving social skills, enhancing social support, increasing opportunities for social contact, and addressing maladaptive social cognition [56].

#### 4.2.2. EI

We found that EI is a possible protective factor against depression, especially in the emotional self-control, problem-solving, peace, and life satisfaction domains.

Emotional self-control includes the ability to process negative thoughts and feelings. Inability to manage negative thoughts causes depression by overstimulating the hypothalamic pituitary adrenal axis and increasing cortisol production. Conversely, a sense of self-control raises self-esteem and decreases depression [57]. Problem-solving skills can buffer the negative correlation between stress and depression. Prior research has shown the effectiveness of problem-solving therapy for preventing and treating depression [58]. Peace, the Eastern concept of happiness, is a well-established protective factor against depression. In addition, showing gratitude can alleviate depressive symptoms [59]. Individual perception of life satisfaction depends not on how much they have but on how they feel. Ways to increase life satisfaction or decrease depression identified in previous studies include optimal self-esteem, good social interactions, and a peaceful environment [58].

Programs for improving EI improving may improve physician wellness and burnout problems [60]. However, no EI improvement programs have been included in Thailand’s conventional medical program. Thus, our study might contribute to a national depression prevention program.

#### 4.2.3. Stressors

Although medical students with depression in this study are twice as likely to be concerned about mental health than individuals without depression, the rate of treatment for depression was relatively low. This finding implies that access to mental health services might be caused not only by low awareness, but also by insufficient knowledge on depression. In addition, having poor mental health impacts their functioning, reducing self-esteem and developing depression. Physical health also affects depression because it affects students’ ability to work or learn effectively [61].

Failure in or obstruction to psychosocial development (romantic relationship problems or difficulty in adjusting social relationships) can cause depression. The impact of age on an individual is discussed in Erik Erikson’s psychosocial development transition periods of “identity vs. role confusion” and “intimacy vs. isolation.” Intimacy not only means romantic relationships, but also friendships and other social relationships. Individuals need much more energy to adjust themselves and progress to the next development stage [62].

Academic boredom might be caused by feelings of anhedonia, leading to depression and, consequently, reduced learning performance and engagement [63]. Some studies found an association of boredom with inattention, anxiety, and depression [64]. Conversely, satisfaction with grades is, unsurprisingly, a possible protective factor against depression.

#### 4.2.4. Demographic Data

As mentioned previously, having a psychiatric or medical illness increases the risk for depression. Medical students with depression showed a significant association between depression and perceived insufficiency of income, although they have a higher actual income per month. As also reported in a previous study, a student’s perception of their financial status, rather than their actual financial status, was significantly associated with depression [65].

#### 4.2.5. Limitations

Our study has some limitations. First, our findings could not explain the causality because of the cross-sectional design. Second, the actual prevalence of depression might be higher than reported because of selection bias. Medical students with severe depression might have been unable to complete the questionnaire because of poor concentration or lack of energy, and thus, might have been excluded from the analysis. However, our results might still be representative of the target population, specifically in northern Thailand, given no significant differences in the percentage of sex and medical students repeating a grade between participants and all medical students. Nevertheless, our findings have limited generalizability to other populations.

## 5. Conclusions

One-fifth of the medical students in this study had depression. Medical students at high risk for depression were those with medical comorbidities, any history of psychiatric disorders, and perceived insufficient income. Medical students facing problems in the following aspects should be monitored for depression: romantic relationships, physical health, mental health, difficulties in social relationships, satisfaction with grades, and boredom with education. Consultants should consider depression in medical students with the following psychiatric comorbidities: anxiety, internet addiction, poor sleep quality, and loneliness. Possible protective factors against depression were EI factors, such as emotional self-control, problem-solving, inner peace, and life satisfaction. Early detection is not enough, and continuous care is crucial. Integrating EI improvement programs might help mitigate depression and should be investigated further.

## Figures and Tables

**Table 1 healthcare-10-00488-t001:** Participants’ demographic data.

Characteristics, *n* (%)	Total (*n* = 706)	Depression (*n* = 149)	Non-Depression (*n* = 557)	*p*-Value
Age (Mean ± SD)	20.6 ± 2.0	20.6 ± 2.0	20.5 ± 2.0	0.622
Female	344 (48.7)	81 (54.4)	263 (47.2)	0.140
Hometown				0.482
Bangkok	58 (8.2)	11 (7.4)	47 (8.4)	
Chiang Mai	213 (30.2)	38 (25.5)	175 (31.4)	
Northern	218 (30.9)	51 (34.2)	167 (30.0)	
Other	217 (30.7)	49 (32.8)	168 (30.2)	
Repeating a grade	23 (3.3)	5 (3.4)	18 (3.2)	1.000
Relationship status				0.159
In relationship	192 (27.2)	47 (31.5)	145 (26.0)	
Single	506 (71.7)	99 (66.4)	407 (73.1)	
Other	8 (1.1)	3 (2.0)	5 (0.9)	
Income per month (Median (IQR))	8000 (6000, 10,000)	9000 (6500, 11,000)	8000 (6000, 10,000)	0.057
Sufficient income	655 (92.8)	130 (87.3)	525 (94.3)	**0.007**
Medical comorbidity	138 (19.5)	42 (28.2)	96 (17.2)	**0.004**
Previous psychiatric disorder	54 (7.6)	23 (15.4)	31 (5.6)	**<0.001**
Alcohol use	397 (56.2)	88 (59.1)	309 (55.5)	0.458
Once a month	303 (41.9)	62 (41.6)	241 (43.3)	
Once a week	94 (13.3)	26 (17.5)	68 (12.2)	
Reason of alcohol drinking				
Socialization	357 (50.6)	74 (49.7)	283 (50.8)	0.438
Stress	57 (8.1)	25 (16.8)	32 (5.8)	**<0.001**
Insomnia	17 (2.4)	9 (6.0)	8 (1.4)	**0.003**
Smoking	33 (4.7)	9 (6.0)	24 (4.3)	0.245
Reason of smoking				
Socialization	17 (2.4)	6 (4.0)	11 (2.0)	0.223
Stress	10 (1.4)	2 (1.3)	8 (1.4)	1.000
Other	6 (0.8)	1 (0.7)	5 (0.9)	1.000
Other substances using	7 (1.0)	2 (1.3)	5 (0.9)	0.454
Psychiatric problem in relatives	39 (5.5)	13 (8.7)	26 (4.7)	**0.047**
Live with				0.201
Parent(s)	524 (74.2)	117 (78.5)	407 (73.1)	
Alone	119 (16.9)	24 (16.1)	95 (17.1)	
Friends/Relatives/Couple	63 (8.9)	8 (5.4)	55 (9.9)	
Living place				0.337
Apartment/Condominium/Rent room	233 (33.0)	51 (34.2)	182 (32.7)	
University dormitory	360 (51.0)	80 (53.7)	280 (50.3)	
House	113 (16.0)	18 (12.1)	95 (17.1)	

SD, standard deviation; IQR, interquartile range. Number in bold are for significant *p*-value at <0.05 (two-sided).

**Table 2 healthcare-10-00488-t002:** Severity of depression according to year of study.

Year of Study	Depression, *n*, (% of Total)	Severity of Depression, *n*, (% of Depression)			Suicidal Ideation in Depression Group
		Mild	Moderate	Severe	
1 (*n* = 195)	42 (21.5)	36 (85.7)	6 (14.3)	0 (0.0)	17 (40.5)
2 (*n* = 105)	27 (25.7)	23 (85.2)	3 (11.1)	1 (3.7)	12 (44.4)
3 (*n* = 112)	16 (14.3)	13 (81.3)	3 (18.8)	0 (0.0)	3 (18.8)
4 (*n* = 120)	29 (24.2)	26 (89.7)	2 (6.9)	1 (3.4)	7 (24.1)
5 (*n* = 95)	21 (22.1)	16 (76.2)	4 (19.0)	1 (4.8)	8 (38.1)
6 (*n* = 79)	14 (17.7)	11 (78.6)	2 (14.3)	1 (7.1)	8 (57.1)
Total (*n* = 706)	149 (21.1)	125 (83.9)	20 (13.4)	4 (2.7)	55 (36.9)

**Table 3 healthcare-10-00488-t003:** Psychiatric problems and possible protective factors correlated with depression.

Factors, *n* (%)/Median (IQR)	Total (*n* = 706)	Depression (*n* = 149)	Non-Depression (*n* = 557)	OR (95% CI)	*p*-Value
Stress due to					
Academic difficulties	592 (83.9)	137 (91.9)	455 (81.7)	2.56 (1.37–4.80)	**0.003**
Financial problems	149 (21.1)	49 (32.9)	100 (18.0)	2.24 (1.49–3.36)	**<0.001**
Relationship with friends	251 (35.6)	79 (53.0)	172 (30.9)	2.53 (1.75–3.65)	**<0.001**
Relationship with one’s beloved	147 (20.8)	53 (35.6)	94 (16.9)	2.72 (1.82–4.07)	**<0.001**
Relationship with family	169 (23.9)	62 (41.6)	107 (19.2)	3.00 (2.03–4.42)	**<0.001**
Relationship with instructor	69 (9.8)	30 (20.1)	39 (7.0)	3.35 (2.00–5.61)	**<0.001**
Physical health	181 (25.6)	68 (45.6)	113 (20.3)	3.30 (2.25–4.84)	**<0.001**
Mental health	205 (29.0)	82 (55.0)	123 (22.1)	4.32 (2.95–6.31)	**<0.001**
Difficulty in social relationship	3 (2,5)	4 (3,6)	3 (2,5)	1.25 (1.15–1.36)	**<0.001**
Satisfy with own grade	7 (5,8)	5 (3,7)	7 (5,8)	0.75 (0.69–0.82)	**<0.001**
Boredom to medical learning	5 (3,7)	7 (5,8)	5 (3,6)	1.44 (1.32–1.58)	**<0.001**
Thinking that wrong decision in studying medicine	69 (9.8)	33 (22.1)	36 (6.5)	4.12 (2.46–6.88)	**<0.001**
Anxiety	38 (5.4)	32 (21.5)	6 (1.1)	25.12 (10.27–61.44)	**<0.001**
Burnout	10 (1.4)	7 (4.7)	3 (0.5)	9.10 (2.33–35.65)	**0.002**
Game addiction	36 (5.1)	14 (9.4)	22 (3.9)	2.52 (1.26–5.06)	**0.009**
Internet addiction	420 (59.5)	112 (75.2)	308 (55.3)	2.45 (1.63–3.68)	**<0.001**
Poor sleep quality	185 (26.2)	74 (49.7)	111 (19.9)	3.96 (2.70–5.81)	**<0.001**
Loneliness	11 (8,15)	16 (12.5,18.5)	10 (8,14)	1.29 (1.23–1.35)	**<0.001**
EI					
Emotional self-control	20 (18,21)	18 (16,20)	20 (19,21)	0.71 (0.65–0.77)	**<0.001**
Empathy	20 (19,22)	19 (17,21)	21 (19,22)	0.79 (0.73–0.85)	**<0.001**
Responsibility	21 (19,22)	20 (18,22)	21 (20,23)	0.83 (0.77–0.90)	**<0.001**
Self-motivation	18 (16,21)	16 (14,18)	19 (17,21)	0.68 (0.63–0.73)	**<0.001**
Problem-solving	17 (15,19)	15 (14,16)	17 (16,19)	0.66 (0.60–0.72)	**<0.001**
Interpersonal relationships	18 (16,20)	16 (14,18)	18 (16,20)	0.81 (0.76–0.87)	**<0.001**
Self-regard	12 (10,13)	10 (9,12)	12 (11,14)	0.64 (0.58–0.70)	**<0.001**
Life satisfaction	20 (17,22)	17 (15,19)	20 (18,22)	0.73 (0.69–0.78)	**<0.001**
Peace	20 (17,22)	17 (15,19)	21 (18,23)	0.70 (0.66–0.75)	**<0.001**
PSS					
Significant others	23 (20,26)	21 (17,24)	24 (21,26)	0.86 (0.82–0.89)	**<0.001**
Family	23 (19,25.3)	20 (16,23.5)	24 (20,26)	0.86 (0.82–0.89)	**<0.001**
Friend	24 (21,26)	21 (18,24)	24 (22,27)	0.86 (0.82–0.89)	**<0.001**
Total	70 (61,77)	62 (52,71)	72 (64,79)	0.94 (0.93–0.96)	**<0.001**

IQR, interquartile range; OR, odds ratio; CI, confidence interval; EE, emotional exhaustion; DP, depersonalization; PE, professional efficacy; EI, emotional intelligence; PSS, perceived social support. Number in bold are for significant *p*-value at <0.05 (two-sided).

**Table 4 healthcare-10-00488-t004:** Results of multivariable binary logistic regression.

Domains	aOR	SE	95% CI	*p*-Value
Model 1: demographic data				
Female	1.43	0.29	0.97,2.12	0.074
Age	0.91	0.05	0.82,1.01	0.079
Repeating the grade	1.04	0.57	0.35,3.06	0.942
Relationship status				
In relationship				
Single	0.80	0.17	0.52,1.23	0.306
Other	1.93	1.63	0.37,10.12	0.435
Sufficient income	0.46	0.15	0.24,0.88	**0.018**
Medical comorbid	1.79	0.40	1.16,2.76	**0.009**
History of psychiatric disorders	3.15	1.01	1.67,5.92	**<0.001**
Alcohol drinking	1.06	0.22	0.71,1.60	0.764
Smoking	1.58	0.71	0.66,3.79	0.304
Other substances using	1.64	1.48	0.28,9.56	0.581
Family psychiatric problem	1.51	0.58	0.71,3.22	0.282
People who live with				
Parent(s)				
Alone	0.87	0.23	0.51,1.48	0.606
Friends/Relatives/Couple	0.50	0.20	0.23,1.10	0.086
Model 2: stressors				
Financial problem	1.28	0.32	0.78,2.08	0.327
Relationship with friends	1.46	0.35	0.92,2.32	0.113
Relationship with one’s beloved	1.80	0.45	1.10,2.94	0.019
Relationship with family	1.49	0.38	0.90,2.45	0.121
Relationship with instructor	1.04	0.34	0.55,1.99	0.900
	1.73	0.41	1.09,2.75	**0.020**
Mental health	2.29	0.54	1.45,3.62	**<0.001**
Difficulty in social relationship	1.13	0.06	1.02,1.25	**0.020**
Satisfy with own grade	0.80	0.04	0.73,0.89	**<0.001**
Boredom to medical learning	1.30	0.07	1.17,1.45	**<0.001**
Thinking that wrong decision in studying medicine	1.50	0.49	0.79,2.85	0.215
Model 3: psychiatric comorbidity				
Anxiety	10.97	5.53	4.09,29.45	**<0.001**
Burnout	4.86	4.08	0.94,25.19	0.060
Game addiction	1.59	0.70	0.67,3.76	0.289
Internet addiction	1.90	0.46	1.18,3.06	**0.009**
Poor sleep quality	2.48	0.57	1.58,3.89	**<0.001**
Loneliness	1.21	0.03	1.15,1.28	**<0.001**
Model 4: EI *				
Emotional self-control (high)	0.46	0.10	0.30,0.71	**<0.001**
Empathy (normal and high)	0.97	0.29	0.54,1.74	0.922
Responsibility (normal and high)	0.69	0.33	0.27,1.76	0.443
Self-motivation (normal and high)	0.86	0.38	0.36,2.05	0.739
Problem-solving (normal and high)	0.31	0.10	0.16,0.59	**<0.001**
Interpersonal relationships (normal and high)	0.57	0.22	0.27,1.21	0.142
Self-regard (normal and high)	0.80	0.28	0.41,1.58	0.523
Life satisfaction (normal and high)	0.46	0.14	0.25,0.85	**0.013**
Peace (normal and high)	0.36	0.13	0.18,0.74	**0.005**
Model 5: PSS				
Significant others	0.96	0.04	0.89,1.05	0.404
Family	0.92	0.04	0.84,1.00	**0.042**
Friend	0.95	0.04	0.88,1.02	0.164
Confounder summary score **				
Demographic	18.00	19.20	2.23,145.57	**0.007**
Stressors	33.58	19.24	10.92,103.25	**<0.001**
Psychiatric comorbidity	28.63	17.28	8.77,93.43	**<0.001**
Emotional intelligence	10.54	6.62	3.08,36.07	**<0.001**
Perceived social support	2.19	2.10	0.33,14.35	0.415

* EI subscales were compared between groups with (1) lower and (2) normal and higher scores. The emotional self-control subscale was grouped into (1) normal and (2) high EI groups because no medical students had low emotional self-control. ** Multivariable logistic regression analysis for risk factors of depression adjusted for confounder summary score. IQR, interquartile range; aOR, adjusted odds ratio; SE, standard error; CI, confidence interval; EI, emotional intelligence; PSS, perceived social support. Number in bold are for significant *p*-value at <0.05 (two-sided).

## Data Availability

All our anonymized dataset files are available from the online database (DOI: 10.6084/m9.figshare.16676725.)

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
