# Peer review of "Prevalence and Associated Factors of Depression in Medical Students in a Northern Thailand University: A Cross-Sectional Study"

_healthcare, 2022, doi:10.3390/healthcare10030488_

Round 1
Reviewer 1 Report
The research topic is very interesting. Sample size is sufficient with a year period of observation for its outcome.
The analysis in the context is abundant which covered many factors. However, the introduction is too short and is encouraged to write more literatures on each / group factors that included in the analysis.
PHQ-9 for assessing depressive symptom that lasts for two weeks, but the assessment duration within the mentioned period in the manuscript is not stated. Perhaps this point can be adding into the methodology.
Besides , criteria for inclusion and exclusion are not details. For instance, why students with historical record of MDD is not excluded? Is it suitable to mix the analysis with samples with MDD and those with screen outcome of depression but has no prior psychiatric record?
Thank you.
Author Response
Response to reviewer
We appreciate your thoughtful evaluation of our work, as well as your pertinent questions and useful ideas. We appreciate your taking the time to read through our article. Your suggestions have resulted in several significant revisions to our manuscript, which we believe have greatly improved it. We've made the following changes to our manuscript in response to your advice:
The reviewers' remarks are listed in the boxes below. The answer to these comments, as well as the changes made to the document, are listed below. The new manuscript also highlights the modifications.
In response to your comment, we will provide an explanation.

Reviewer 2 Report
The purpose of this study was to investigate the prevalence and associated factors of depression in medical students. As we know, the association between depress, anxiety, poor sleep quality, and loneliness have been well documented in mental health filed. For this reason, overall there is no rational of the importance of this study in mental health filed.
In addition, this introduction part is inadequate of review, this study also has serious flaws in terms of rationale, methods, and development of preventive interventions programs discussed in the manuscript.
Major concern:
Introduction
Line 33
Please add literature reviews for prevalence of depression of medical students in Thailand. à ref. 12
Line 34-37
Also, this manuscript has inadequate of evidences about depression care of medical students.
Currently, is there any preventive intervention program?
Methods section
Line 73-79
Why included General Demographic Data ? for example, hometown… you need to add reference.
Discussion section
Line 242-254
A review of the EI program with interventions for students with depression is needed.
You need to add reference.
Conclusions section
Line 290-294
You need to modify the conclusion part to fit your study purpose and study design.
- your primary objective is to investigate depression prevalence; the secondary objective is to explore both potential risk and protective factors of depression among Northern Thai medical students, including all years of study.
Other
I recommend that this manuscript should be edited by an English professional editor for more readable. There are several grammatical errors.
Round 2
Reviewer 2 Report
I would like to thank the authors for their work on this manuscript.